# Electrically Conductive MXene-Coated Glass Fibers for Damage Monitoring in Fiber-Reinforced Composites

**Christine B. Hatter [1]** , **Asia Sarycheva [1]** , **Ariana Levitt [1]** , **Babak Anasori [2]** , **Latha Nataraj [3]** and **Yury Gogotsi [1,\***

[1] A.J. Drexel Nanomaterials Institute and Department of Materials Science and Engineering, Drexel University, 3141 Chestnut St., Philadelphia, PA 19104, USA; CBH52@drexel.edu (C.B.H.); as4357@drexel.edu (A.S.); asl95@drexel.edu (A.L.)

[2] Integrated Nanosystems Development Institute and Department of Mechanical and Energy Engineering, Indiana University-Purdue University Indianapolis, 723 W. Michigan St, Indianapolis, IN 46202, USA; banasori@iupui.edu

[3] U.S. Army Research Laboratory, Aberdeen Proving Ground, MD 21005-5069, USA; latha.nataraj.civ@mail.mil

\* Correspondence: Gogotsi@drexel.edu

**Abstract:** Multifunctional fiber-reinforced composites play a significant role in advanced aerospace and military applications due to their high strength and toughness resulting in superior damage tolerance. However, early detection of structural changes prior to visible damage is critical for extending the lifetime of the part. MXenes, an emerging class of two-dimensional (2D) nanomaterials, possess hydrophilic surfaces, high electrical conductivity and mechanical properties that can potentially be used to identify damage within fiber-reinforced composites. In this work, conductive $Ti_3C_2T_x$ MXene flakes were successfully transferred onto insulating glass fibers via oxygen plasma treatment improving adhesion. Increasing plasma treatment power, time and coating layers lead to a decrease in electrical resistance of MXene-coated fibers. Optimized uniformity was achieved using an alternating coating approach with smaller flakes helping initiate and facilitate adhesion of larger flakes. Tensile testing with in-situ electrical resistance tracking showed resistances as low as 1.8 kΩ for small-large flake-coated fiber bundles before the break. Increased resistance was observed during testing, but due to good adhesion between the fiber and MXene, most connective pathways within fiber bundles remained intact until fiber bundles were completely separated. These results demonstrate a potential use of MXene-coated glass fibers in damage-sensing polymer-matrix composites.

**Keywords:** MXenes; damage-sensing composites; multifunctional polymer composites; glass fiber coatings; fiber reinforced composites

## 1. Introduction

Fiber-reinforced composite systems play a significant role in modern structural designs found in aerospace and military applications due to their high mechanical strength and lightweight properties. Over the last four decades, aircraft designs have seen a 50% increase in the use of composite materials resulting in a growing need for better understanding of various types of damage to these parts as well as methods for monitoring them [1]. However, common defects that arise during manufacturing, such as porosity or voids within the polymer matrix, can be caused by non-optimal curing parameters or inclusion of foreign bodies such as leftover backing film or greasy finger marks [2]. Structural parts already in-service suffer from defects arising from normal operational wear or unexpected events such as impact from foreign objects resulting in delamination (separation of layers within the laminate

structure), matrix cracking, fiber-matrix debonding and fiber breakage leading to significant loss in mechanical properties [3–6]. As a result, predicting damage to a part prior to mechanical failure is needed to optimize the life of the composite and avoid a catastrophic failure in service [7]. Previous works have explored incorporating nanomaterials as additional fillers to take advantage of their ability to improve interfacial strength as well as damage-sensing capabilities [7–12]. Gao et al. showed that coating glass fibers with multiwalled carbon nanotubes (MWCNTs) produced semiconducting glass fibers with resistances ranging from 104 to 107 Ω [13]. Additionally, further studies by Gao's group showed full dispersion of carbon nanotubes (CNTs) into woven glass fiber/epoxy composites and successful monitoring of changes in electrical resistance during repeated impact tests showing upwards of 1600% change in resistance with increasing damage area [14]. However, due to the inert nature of the surface of CNTs and MWCNTs, additional steps must be taken to disperse them into polymer systems or attach them to glass fiber surfaces.

Graphene, the two-dimensional (2D) material that was isolated in 2004, has also been implemented into polymer composites systems owing to its high mechanical and electrical properties [15,16]. Wang et. al. showed that the addition of 0.5 wt.% graphene to glass fiber-epoxy composites improved both the impact modulus and flexural modulus of the overall composite; however, electrical properties were not reported [17]. A later study showed that interlaminar shear strength can be improved from 25.6 to 36.23 MPa by varying solution concentration of graphene oxide and by covalently bonding graphene oxide onto amine-functionalized glass fiber surfaces for solution concentrations of 1.5 mg/mL [18]. The use of graphene for damage-monitoring has also been extended to carbon fiber-reinforced composites. Du et al. successfully introduced thermally reduced graphene sheets into carbon fiber-reinforced epoxy laminates with normalized resistance change. In this study, resistance increased monotonously with crack growth within the samples [19]. Graphene derivatives have provided an easier route for incorporating carbon-based nanomaterials into both polymer-matrix and fiber-reinforced polymer systems. However, graphene derivatives have significantly lower electrical conductivity compared to pristine graphene, thus creating limitations [20].

MXenes have emerged as 2D alternatives to graphene for use in composite systems with over 30 different synthesized stoichiometric compositions to date [21–23]. Derived from MAX phases, MXene's structure is $M_{n+1}X_nT_x$, where M is an early transition metal such as Ti, V, Nb, and Mo, X is carbon and/or nitrogen, $n$ is equal to 1–4. $T_x$ represents surface functional groups such as -O, -OH, or -F [24,25] resulting from wet chemical synthesis and making the surface of MXene hydrophilic. $Ti_3C_2T_x$ MXene possesses electrical conductivities upwards of 10,000 S/cm and the highest Young's modulus, 330 GPa, for solution processed 2D materials [26–28]. Due to the oxygen containing functional groups, MXenes can be easily dispersed into polar organic solvents and mixed with a variety of polymers [29–33]. Previous studies have shown the potential of MXene composites for achieving outstanding electromagnetic interference (EMI) shielding, gas and water separation membranes and flexible electrodes [34–37]. However, there are limited studies examining MXenes in fiber-reinforced composites. A recent study introduced MXenes into carbon-fiber composites for improved wettability between carbon fibers and the surrounding epoxy matrix [38]. However, the unique combination of electrical and mechanical properties of MXenes was not fully utilized in this work. There are no reports on glass fibers coated with MXene or use of MXenes in manufacturing damage-sensing composites.

Here, we present the first study of manufacturing electrically-conductive glass fibers through a simple dip-coating process using a colloidal solution of $Ti_3C_2T_x$ MXene. MXene sheets were attached to surface-treated glass fibers providing a uniform coating and transferring electronic properties of $Ti_3C_2T_x$ to the insulating fibers. Electrical resistance monitoring was performed in-situ with tensile measurements on fiber bundles to assess potential tracking of mechanical failure in epoxy composites.

## 2. Materials and Methods

### 2.1. Synthesis of Ti$_3$C$_2$T$_x$ MXene

Ti$_3$C$_2$T$_x$ MXene was prepared by etching Ti$_3$AlC$_2$ MAX phase powder (Carbon-Ukraine, Kyiv, Ukraine) [24,39]. First, 2 g of lithium fluoride (LiF, Alfa Aesar, Haverhill, Massachusetts, USA) was dissolved in 20 mL of 9M hydrochloric acid (HCl, Fisher Chemical, Waltham, Massachusetts, USA) solution and stirred for 5 min at room temperature. Next, 2 g of Ti$_3$AlC$_2$ MAX powder was added slowly over 10 min to the HCl-LiF solution and the reaction was stirred for 24 h at 35 °C. The etched mixture was then washed several times with deionized (DI) water by centrifugation at 3500 rpm until pH 6–7 was reached. To obtain large Ti$_3$C$_2$T$_x$ flakes, additional washes were performed past reaching neutral pH and the dark supernatant containing large, delaminated Ti$_3$C$_2$T$_x$ was collected. For smaller flakes, the washed sediment was redispersed into 20–30 mL of DI and bath sonicated for 30 min under Ar bubbling followed by centrifugation at 3500 rpm for 1 h. The dark supernatant containing small delaminated flakes was then collected. Sonication yielded small flakes with lateral sizes of 105–795 nm and 1–5 µm for larger flakes. Flake size measurements were taken from 15–20 images collected via transmission electron microscopy and averaged. Freestanding films were prepared from solutions containing small and large flakes via vacuum-assisted filtration and the electrical conductivity of the films was measured using a 4-point probe setup (Jandel ResTest).

### 2.2. Surface Treatment of S-Glass Fibers

To improve MXene flake adhesion and electrical properties of glass fibers, fiber surfaces were chemically treated and exposed to oxygen plasma. As-received S-glass fibers (provided by Army Research Laboratory, Aberdeen Proving Ground, MD, USA) were heat treated at 600 °C for 1 h in a box furnace (MTI KSL 1100X) to remove any residual coating on fibers from the commercial manufacturing process. Next, fibers were treated with 3:1 weight ratio of sulfuric acid and hydrogen peroxide (Piranha) solution for 10 min followed by washing with DI water 5 times until neutral pH was reached. Glass fibers were then exposed to oxygen plasma treatment (Tergeo Plus, PIE Scientific, San Francisco, CA, USA) at 50 and 150 W power for different durations (0.5, 2 and 5 min). For in-situ resistance testing, only samples treated with 50 W power for 5 min were used due to lowest electrical resistance properties.

### 2.3. Dip Coating of Glass Fibers with MXene

To coat the glass fibers with MXene, 10 mL of aqueous Ti$_3$C$_2$T$_x$ MXene solution (3–5 mg/mL) was poured into a plastic weighing boat (10 cm diameter). Next, surface treated glass fiber bundles were dipped into a MXene solution containing large flakes for 5 min followed by drying in air for 5 min. This process was repeated with varying number of dips from 5 to 20 and 50 and electrical resistances were measured. Fibers coated with large flakes (LF) were dried at 200 °C for about 15 h under vacuum to remove water and ensure full adhesion of MXene flakes to glass fiber surface.

An alternative coating method was explored utilizing alternating coatings of small and large MXene flakes for improved coverage and electrical resistance. For this process, surface treated glass fibers were dip-coated in MXene solution of small flakes for 5 min then allowed to dry (5 min). Next, the fibers were dipped into a large flake solution following the same process to obtain an inner coating of small flakes secured by larger flakes. This small-large flake (SLF) coating procedure was repeated 5 times for each coating of small and large flakes. Lastly, SLF-coated fibers were dried at 200 °C for about 15 h under vacuum to remove water and ensure full adhesion of MXene flakes to glass fiber surface.

### 2.4. Tensile and In-Situ Electrical Resistance Measurements

Ti$_3$C$_2$T$_x$-coated fiber bundles were cut into uniform lengths of 3 cm and diameters of 0.5 mm. Paper tensile frames (length 2 cm × width 1.5 cm) were prepared with copper tape at the ends for

electrical measurements. Fibers were mounted onto the frames using conductive silver paste and allowed to dry in air at room temperature for 2 h. Preliminary resistance measurements were taken prior to testing to ensure that conductive pathways were not disrupted during mounting. Samples were loaded into a tensile tester (Instron 3365) and connected to a digital multimeter (Keysight 34461A digital multimeter) for measuring electrical resistance. The electrical resistance was allowed to stabilize for 1 min before starting the tensile test. Tensile testing was performed at a rate of 1 mm/min using a 100 N load cell. The resistance of the fiber bundle was tracked until the break point and/or till the limit of the resistance tracking was reached.

*2.5. Characterization*

Images of small and large flakes were collected using a transmission electron microscope (TEM, JEOL 2100) using an accelerating voltage of 200 kV. MXene solutions were drop-casted onto 3 mm diameter lacey carbon TEM grids for analysis. Surface morphologies of glass fibers and MXene-coated fibers were observed with a scanning electron microscope (SEM) (Zeiss Supra 50 VP). Samples were sputter-coated with platinum/palladium for 40 s prior to imaging. Raman spectra were recorded with a Renishaw InVia microscope in an inverted reflection mode (Gloucestershire, UK), equipped with a 63x (NA = 0.7) objective. The laser line used was 785 nm (solid state diode laser) with a 1200 line/mm grating. Spectra were collected with 5% laser (1.6 mW/cm$^2$) power during 120 s.

## 3. Results and Discussion

A schematic of the surface treatment and dip-coating processes of glass fibers is shown in Figure 1. For best adhesion, oxygen plasma were utilized to create a hydrophilic surface after removal of residual sizing from manufacturing. Coating processes were repeated until uniform MXene coatings were obtained along the entire length of the fibers. In the SLF coating method, the use of small flakes also creates an initiation layer for larger flakes as well as ensuring complete coverage of fiber surfaces within dip-coated bundles. Removal of excess water molecules from between MXene flakes was achieved through simple heating under vacuum.

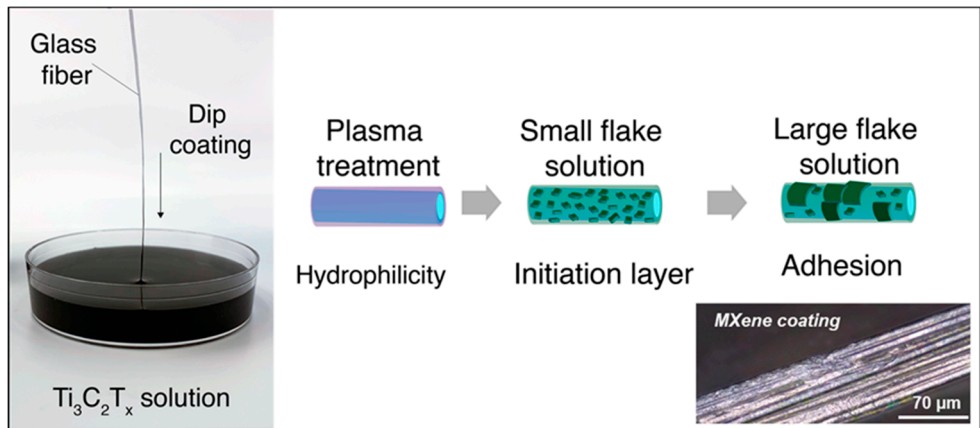

**Figure 1.** Schematic of the dip-coating process for coating $Ti_3C_2T_x$ MXene onto glass fibers. Before dip-coating, fibers were treated with oxygen plasma to add a hydrophilic layer for improved adhesion. Fiber bundles were coated with MXene solutions using two methods: (1) large flakes only and (2) alternating small flake-large flake approach using separate solutions of small and large MXene flakes. Optical image of small-large flake (SLF)-coated fiber bundle post heat treatment.

Electrical properties of the MXene-coated fibers were directly related to the flake size, coating uniformity, electrical conductivity and adhesion to fiber surface. To evaluate the effects of MXene flake size, films were made from small and large flakes using vacuum-assisted filtration and their Raman spectra were recorded (Figure 2a). The characteristic $Ti_3C_2T_x$ peaks can be seen in the region

between 100 and 800 cm$^{-1}$. Both spectra agree with previously reported Raman spectra for Ti$_3$C$_2$T$_x$ [40], which indicate no observable oxidation or degradation happened during the sonication and coating process. The peak at 123 cm$^{-1}$ is a typical resonant Ti$_3$C$_2$T$_x$ peak and the distinct peak at 202 cm$^{-1}$ is attributed to out-of-plane vibrations of Ti, C and surface groups' atoms. Peaks found in the range of 200–210 cm$^{-1}$ typically correspond to A$_{1g}$ vibrations of Ti, C and O atoms in Ti$_3$C$_2$O$_2$ and Ti$_3$C$_2$O(OH). E$_g$ and A$_{1g}$ vibration modes for OH group at 268–286 cm$^{-1}$ and 504–520 cm$^{-1}$, respectively, confirm the presence of hydroxyl functionalities on the MXene surface [41]. For both large (no sonication) and small (sonicated) Ti$_3$C$_2$T$_x$ flakes, oxygen (O) and OH terminations provide the hydrophilic properties needed for improved adhesion to the treated glass fiber surface. In the case of small flakes, the out-of-plane peak related to vibrations of Ti, C and surface groups atoms is less prominent, which could be related to increased defects in the film made of smaller flakes. Corresponding TEM images (Figure 2b,c) show varying lateral flake sizes of both small and large flakes and diffraction patterns. Larger flakes were observed to have electrical conductivity of ~3500 S/cm when freestanding films were prepared, while smaller flakes showed conductivity of 2200 S/cm in a film. Prior studies have shown a direct relationship between MXene flake size and electronic properties of resultant films [42].

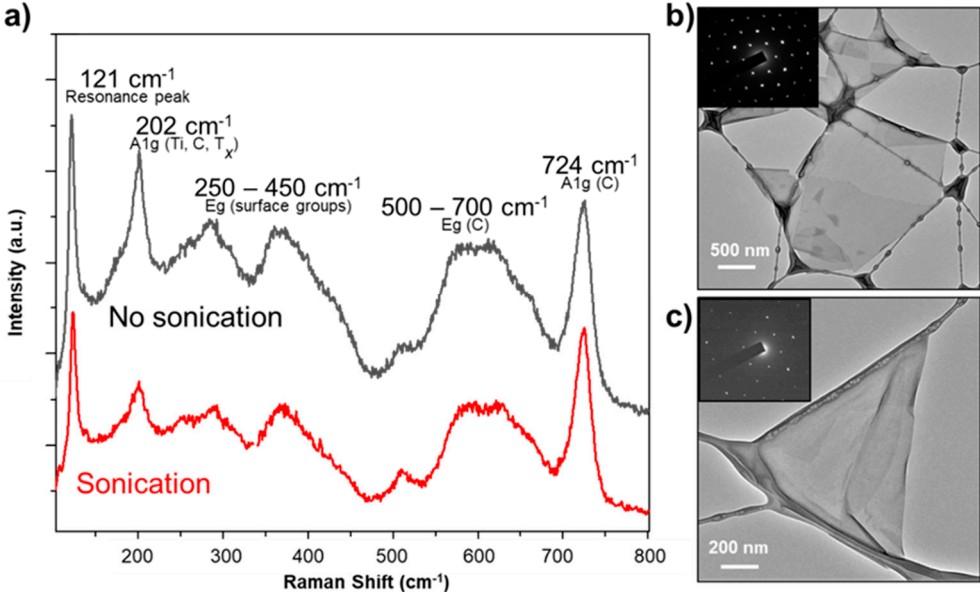

**Figure 2.** Raman spectra (**a**) and TEM images of single MXene flakes of different sizes (**b**,**c**). Large Ti$_3$C$_2$T$_x$ flakes (**b**) were collected post-etching and smaller flakes (**c**) were produced using bath sonication. Insets show selected area diffraction patterns for single-crystal MXene flakes.

After dip-coating and drying fiber bundles, Raman spectra were collected to assess the uniformity of the MXene coating. For each fiber bundle, three different locations were investigated, and Raman spectra were collected every 1 μm over a 10 μm distance. Figure 3a shows averaged spectra for both 5-dip LF and SLF-coated fibers. Presence of Ti$_3$C$_2$T$_x$ peaks are observed for all dip-coated fiber bundles with small shifts to lower wavenumbers. Additionally, SLF coating method exhibited an increase in intensity of MXene resonance peak as well as A$_{1g}$ (Ti, C, T$_x$) peaks corresponding to highly aligned MXene flakes on the fiber surface. SEM images of fiber surfaces revealed good adhesion and coating of Ti$_3$C$_2$T$_x$ for both LF MXene coating methods (Figure 3b). MXene flakes can be seen covering entire length of fibers with no glass surface visible. Best MXene coatings were obtained after high-temperature heat treatment and Piranha solution removing possible sizing from manufacturing.

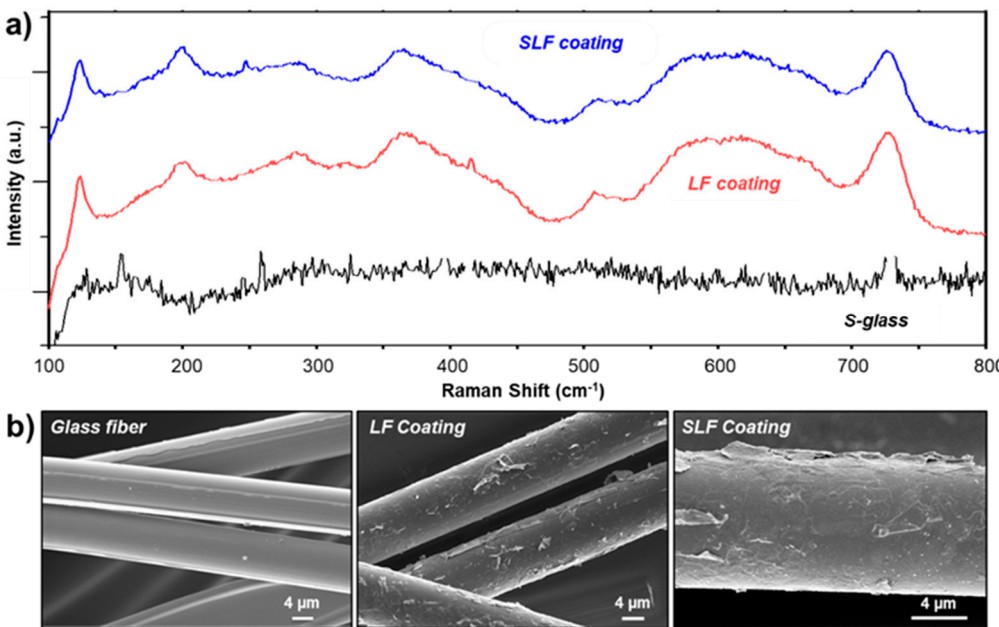

**Figure 3.** Raman spectra of pure S-glass and MXene dip-coated fibers with corresponding $Ti_3C_2T_x$ peaks present after dip-coating process (**a**). SEM images of fiber surfaces for oxygen treated S-glass fibers and dried MXene-coated fibers (**b**).

To assess extent of coating uniformity and its effect on electronic properties, glass fibers were surface treated with varying oxygen plasma treatment parameters. The addition of an oxygen-rich hydrophilic layer to glass fibers allows for best adhesion of $Ti_3C_2T_x$ MXene providing optimal coating. Figure 4a,b show SEM images of plasma-treated fiber fracture surfaces at different plasma power intensities (50 and 150 W) after five dips in aqueous MXene dispersion of large flakes. MXene coverage is achieved in both systems with visible $Ti_3C_2T_x$ flakes adhered to fiber surfaces; however, at higher intensity a thicker and less uniform MXene coating is obtained. At 150 W, more MXene adheres to glass fiber surface, but the coating is less uniform and contains visible aggregates of large $Ti_3C_2T_x$ flakes. Corresponding electrical resistance plots of different plasma power and treatment times are shown in Figure 4c,d. With increased MXene adhesion, fiber bundles treated at 150 W power and maximum of 50 dips showed lowest resistance properties ranging from 2.5 +/− 0.47 kΩ compared to 50 W which showed slightly higher resistance values averaging 14 +/− 9.6 kΩ (Figure 4c,d). At higher plasma power, the induced hydrophilic surface of the treated glass fibers allows for better interaction with oxygen containing functional groups on the MXene surface [43]. For dip-coated fibers treated at 50 W, a slight increase in electrical resistance was observed above five dips (for 20 and 50 dips); however, overall resistance never exceeded 110 kΩ. At lower plasma power, there is likely less coverage of oxygen groups on the surface of glass fibers, compared to 150 W, resulting in a decrease in the number of MXene flakes that can adhere to the glass surface and participate in electronic pathways. When comparing treatment times for both 50 and 150 W, only small changes in electrical resistance were observed when increasing from 0.5 to 5 min plasma exposure (25 kΩ or less). Fibers treated with 150 W power all displayed lower electrical resistances due to hydrophilicity of the glass fibers allowing for maximum coverage with MXene. MXene-coated fibers exhibiting insufficient coating uniformity can result in poor damage-monitoring capabilities due to easy removal during testing. As a result, glass fibers treated at 50 W power for 5 min were used for assessing damage-monitoring capabilities of MXene-coated fibers.

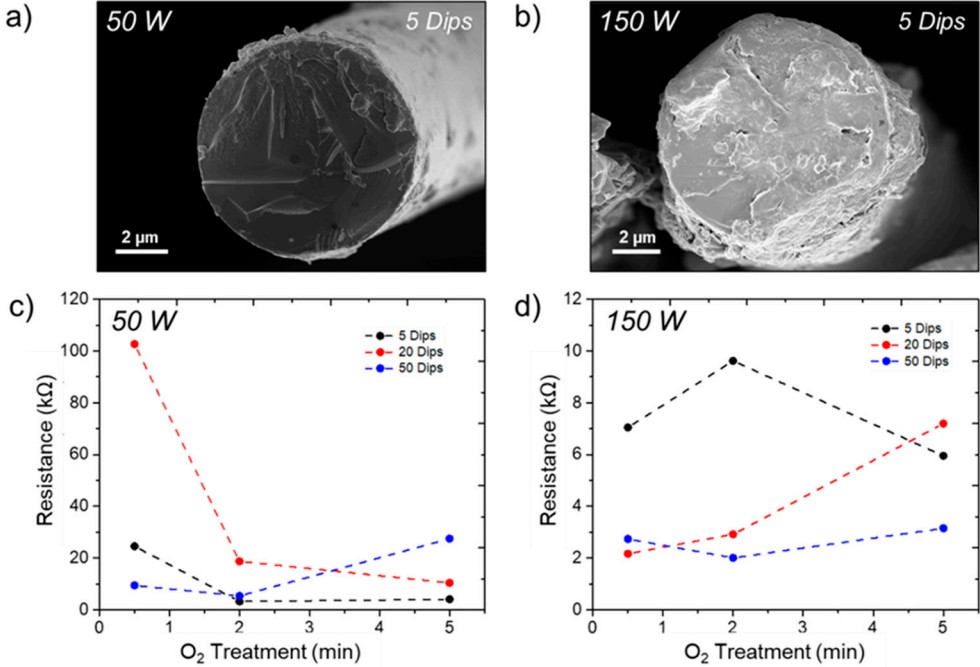

**Figure 4.** Comparison of MXene coatings based on oxygen plasma surface treatment and corresponding resistance values. SEM images of $Ti_3C_2T_x$-coated fibers using 50 W (**a**) and 150 W (**b**) power during plasma treatment. Resistance values for varying surface treatment times are provided in (**c,d**). Lines connecting points in (**c,d**) are a guide for the eye.

For damage-monitoring in fiber-reinforced composites, electrically conductive fibers are essential for determining the point of fracture within the system based on changes in electrical properties. Therefore, continuous conductive pathways along the fibers are necessary. Electrical properties of MXene flakes can be utilized for tracking deformation and fracture of glass fibers within a polymer matrix via electrical resistance. Additionally, the simple dip-coating method eliminates extra processing steps and can be embedded directly into the polymer after drying. In-situ electrical resistance monitoring was performed on MXene-coated glass fibers while in tension utilizing LF and SLF coating methods. Due to overall coating uniformity, fibers treated under 50 W power for 5 min in oxygen plasma were prepared prior to dip-coating with MXene. The SLF dip-coating method utilizes small flakes to penetrate fiber bundles ensuring all fiber surfaces are covered while large flakes further increase electrical conductivity and secure smaller flakes to the fibers. Figure 5a,d shows resistance changes with increasing tensile load as a function of time for LF- and SLF-coated glass fiber bundles. For this study, engineering stress-strain curves were not utilized for determining mechanical properties of MXene-coated fibers. Since bundles with varying number of fibers were investigated, exact information for influence of MXene coating cannot be accurately assessed. As fiber bundles are strained, an initial decrease in electrical resistance is observed as bundles are stretched until taught. This decrease is due to complete alignment of fibers within the stretched bundle and full contact creating a complete conductive pathway between all fibers. As a result, both five-dips LF- and SLF-coated fiber bundles reach low resistance values of 3 kΩ and 1.8 kΩ, respectively. The SLF method exhibited the lowest resistance values even when compared to large flake coatings using 20 and 50 dips confirming small flakes can penetrate into bundles for improved coverage and overall electrical properties.

MXene-coated fiber bundles were strained until break and subsequent loss of electrical resistance monitoring. As tensile load approaches 5–6 N near break after ~275 s, an increase in electrical resistance is observed as individual fibers begin to break within the bundle and begin to separate disrupting the conductive pathways (Figure 5a,d). After tensile load dropped to 0 N at break (Figure 5b,e), no distinct fracture point was observed in the fiber bundles however electrical properties remained. Since individual fiber breakage occurs at different locations within the bundle, some conductive

pathways remained after a drop in load was observed however resistance values did increase. Limit of resistance tracking (~1 GΩ) did not occur simultaneously with tensile break but only after all connective pathways within the fiber bundle were lost. This event occurred after allowing fiber bundles to be continuously pulled past the tensile break point over several minutes until the two halves of the bundle were completely separated.

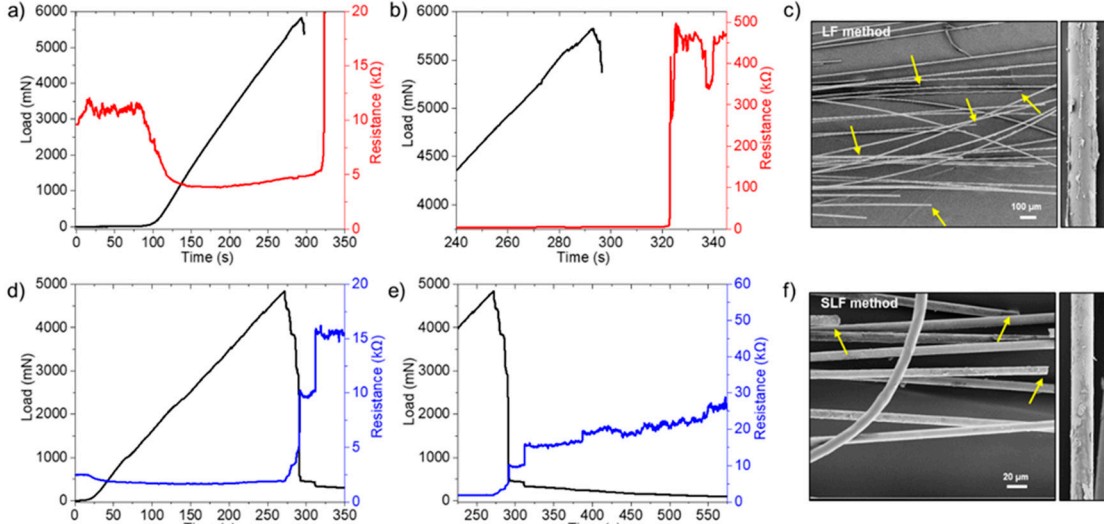

**Figure 5.** Electrical resistance monitoring via tensile strain measurements over time for LF (**a**,**b**) and SLF (**d**,**e**) MXene dip-coating methods. Plots in (**b**,**e**) show resistance behavior around fiber bundle fracture prior to resistance monitoring limit. Corresponding SEM images of fiber bundles after break and single fiber surfaces post-testing (**c**,**f**). SEM images of single fiber surfaces show MXene coating remains on the fiber surface after testing.

LF-coated fibers using five dips showed a sharp increase in electrical resistance shortly after tensile break followed by a brief stabilization around 450 kΩ (Figure 5b) before rapidly increasing to the resistance limit. SLF coating method showed a two-step change in resistance from ~10 to 15 kΩ followed by a steady increase in electrical resistance. However, more time during testing was needed to reach the resistance limit of the instrument (Figure 5e). SLF-coated fibers showed lower resistance values even after fracture due to better coating resulting from the use of both small and large MXene flakes. As the fiber bundle reaches break, conductive pathways from outer shell large flakes are first to be disrupted; however, conductive pathways from smaller flakes remain intact longer as bundles are pulled apart. SEM images of broken fiber bundles (Figure 5c,f) show individual fibers breaking at various locations (yellow arrows), causing increases in resistance as bundles are slowly pulled apart. Additionally, SEM images of single fiber surfaces show loss of MXene coating due to friction from fibers sliding past one another during testing which adds to the increased electrical resistance observed. It is worth mentioning that fiber breakage within a composite system using DC measurements offers one failure mode within fiber-reinforced systems; however, to assess other types of failures, such as debonding or delamination, AC measurements can also be implemented [44–46]. Prior studies have shown MXenes can be incorporated into polymer matrices producing electrically conductive composites. Addition of MXene-coated fibers will improve overall conductivity and AC-based detections.

Lastly, the extent of damage to MXene coating after tensile fracture was explored via Raman spectroscopy. Figure 6a shows an optical image of a broken fiber from a SLF dip-coated fiber bundle after 30 days. It can be seen the MXene coating remains intact along the entire length of the glass fiber. Raman spectra were collected at various locations along the coated fiber starting at the point of break, 5 μm and 15 μm (Figure 6b). $Ti_3C_2T_x$ peaks are clearly visible indicating that even though some layers may have been lost during testing, the SLF coating method provides complete coverage of glass fibers

as well as good adhesion after an extended period of time. Minimal oxidation of the MXene-coated fibers occurred as no photoluminescence background nor $TiO_2$ peaks are observed [47]. Carbon-related peaks are observed at higher wave numbers; however, these peaks are observed in pristine fibers as well. Transfer of electronic properties show the potential practical use of these fibers as woven fabrics embedded in polymer matrix systems. $Ti_3C_2T_x$ MXene has been shown to possess higher electrical conductivity compared to solution processed MWCNTs, graphene oxide and reduced graphene oxide with simpler processing. Furthermore, prior studies have shown the excellent electronic properties of $Ti_3C_2T_x$ can also impart EMI shielding capabilities [48] which can be an additional property in conjunction with damage-monitoring. Lastly, $Ti_3C_2T_x$ MXene has additional attractive properties such as tunable plasmonic effects which enable light-to-heat conversion [49] fostering self-healing possibilities within composites.

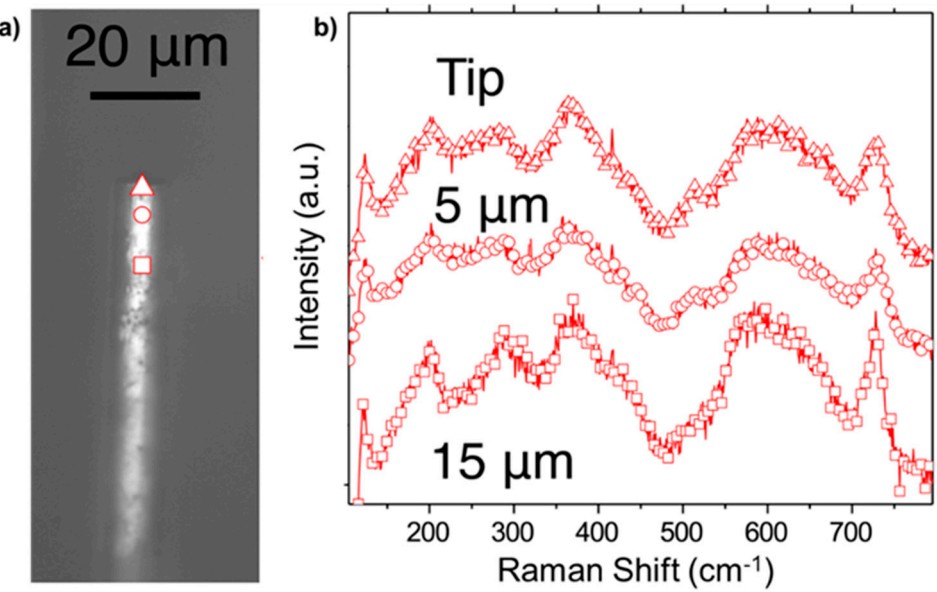

**Figure 6.** Optical image of a single SLF MXene-coated fiber within a broken bundle (**a**). Raman spectra of broken fiber post-tensile testing from break point to 15 μm along the fiber length (**b**). The SLF coating method shows clear MXene peaks remain after break with no presence of oxidation ($TiO_2$) after tensile strain.

## 4. Conclusions

$Ti_3C_2T_x$ MXene-coated glass fibers were produced through a facile dip-coating process following oxygen plasma treatment of glass fiber surfaces for improved MXene adhesion. Addition of the hydrophilic layer improved $Ti_3C_2T_x$ coatings, imparting electrical properties to insulating glass fibers. Increasing plasma treatment power to 150 W and number of dips to 50 provided a fiber bundle resistivity of 1.3 Ω*cm; however, coating uniformity was sacrificed due to agglomeration of MXene flakes on the surface. Uniform MXene coverage was achieved with surface treatments at 5 W and 5 dips exhibiting the most uniform coverage and electrical resistances as low as 5 kΩ prior to tensile testing. The lowest resistivity value reached was by using alternating small and large flake coatings (1.18 Ω·cm) when all connective pathways within the fiber bundles were in contact during in-situ resistance testing. When individual fibers were broken and the fiber bundle was completely separated, a gradual increase in resistance was measured. MXene coatings were partially lost during tensile testing due to friction; however, electrical resistance was still measurable until complete separation of fiber bundles. This study provides a simple route for imparting electronic properties of MXenes to insulating fibers for incorporation into polymer or ceramic matrices and developing damage-monitoring composite systems.

**Author Contributions:** Conceptualization, C.B.H., L.N.; methodology, C.B.H.; validation, C.B.H., A.S., A.L.; investigation, C.B.H., A.S., A.L.; writing—review and editing, C.B.H., A.S., A.L., B.A., L.N., Y.G.; visualization, C.B.H., A.S.; supervision, L.N. and Y.G. All authors have read and agreed to the published version of the manuscript.

**Funding:** This research has been possible with the support of a collaborative research agreement between the U.S. Army Research Laboratory and Drexel University (W911NF-17-2-0228).

**Acknowledgments:** The authors would like to thank Christopher Li for use of tensile testing equipment. A.L. was supported by the National Science Foundation Graduate Research Fellowship under Grant No. DGE-1646737. SEM and TEM analyses were performed at the Core Research Facilities (CRF) at Drexel University.

**Conflicts of Interest:** The authors declare no conflict of interest.

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
