# Peer review of "Electrically Conductive MXene-Coated Glass Fibers for Damage Monitoring in Fiber-Reinforced Composites"

_carbon, 2019_

Round 1
Reviewer 1 Report
In attached file please find the issues to be considered before publishing final version of this manuscript.

Author Response
Response to comments of the reviewers
Dear Dr. Zhang,
We are grateful to the reviewers for the valuable comments. Below we address all points raised by the reviewers. We have added additional context and revised our paper accordingly. All changes have been clearly marked in the manuscript. Hopefully this revision has rendered the paper acceptable for publication.
The revised manuscript has been checked and approved by all co-authors.
On behalf of all authors,
Yury Gogotsi
- It should be carefully checked to keep all structural formulas similar i.e.: L65 Mn+1XnTx (Mn+1XnTx), L288/301 TiO2 (TiO2)
Response: We thank the reviewers for their thorough review. We have modified the text to align all structural formulas.
- L119/127 -> “fibers were dried at 200°C overnight (15 h) under vacuum” Except Polar Nights, normal night takes roughly 6-10 hours. Thus, please remove overnight from both sentences. .
Response: We have modified the text accordingly.
- According to Reviever’s experience - while conducting the Raman measurements on Ti3C2Tx MXenes, with identical equipment and setup - the 5% of 785 nm laser power is enough to harm the MXene sample. It is clearly visible as a rise of a D - G peaks in the range of 1200 - 1750 cm-1, related to the presence of amorphous carbon, and/or carbidederived carbon. Those part of spectra are not presented. It would be wise to prove that chosen parameters are adequate for the measurements. Raman spectra present nice features in the range of 100-800 cm-1. However, the description of observed peaks could be upgraded. Authors should familiarize with the work of Scheibe et al. (Appl. Surf. Sci. 479 (2019) 216-224, especially supplementary information, where predicted vibrations by cited manuscript, were experimentally assigned to specific surface functional groups. Those part could be written with more details, especially when MXene’s functional groups take active part during binding to the plasma treated glass fibres. TEM images contain interesting SAED pattern insets. However, it would be also good to place there histograms related to flake size distributions described in L97/98. In the case of SEM (Fig 3.) it would be good to perform the EDS mapping to prove the full coverage of the fiber with MXene flakes. (IF POSSIBLE). The Raman analysis again show only half spectra. The TiO2 anatase peak at 150 cm-1 is actually present in highly oxidized MXene samples with TiO2 grains. The oxidation of MXenes usually is more subtle and can be observed by rise of D band area. Therefore, please prove that MXenes did not oxidize via full Raman spectra, or remove the following sentence: “Furthermore, the absence of anatase TiO2 peak at 150 cm-1 reveals the MXene coating does not oxidize while adhered to glass fiber surfaces even after an extended period of time. Limited oxidation of the MXene-coated fibers…”
Response: We have provided more clarification of Raman peaks and included more details in the manuscript and modified the manuscript. Authors did not include D and G band region of Raman spectra as D and G bands are present in pristine fibers as well. Please see the spectrum below: (Please see attached)
- This work lacks comparison with existing, carbon based materials that has been widely used in PMC. CNT may be a bit challenging to disperse in polymer systems, but graphene should have less concern. The authors are suggested to include and compare the literature data, both on the aspect of strength enhancement detection (e.g. resistance change). If experimental data is available, it is even better as a fair reference to the MXene. Other factors like higher density/weight for MXene vs. carbon based material should also be included.
Response: We have modified the text accordingly.
- The study seems to be limited to unidirectional composites, and the testing regimen is limited only to (DC) resistance. Better consideration on real PMC and FRP architecture should also be commented. Apart from unidirectional types, it is known that PMC and FRP structure may also take the form of cross-ply and 2D or 3D woven fibres. The authors are suggested to show, either by measuring or explaining how simple measurements on unidirectional composite can be related to these real FRP architecture. One main reason for this question is due to the known electrical anisptropy (e.g. between the through-plane and in-plane) in FRP. (see e.g. Journal of Composite Materials. 2020;54(7):867-882.) While DC measurement is OK for fibre breakage detection (see e.g. Composites: Part B 42 (2011) 77–86), fibre breakage is just one of possible failure modes in FRP. AC is more useful for other types of defects like delamination or debonding (see e.g. Compos Sci Technol 2001;61:855–64.). The authors should discuss what is the potential effect of using MXenes-based composite AC based detection is used, and if this is going to be beneficial or not.
Response: We have taken these works into consideration and modified the text.
- The measurements shown in Figure 5 focuses only to study inelastic deformation related damage. However, better wear/failure prediction may be done if information of the composites behaviour in the elastic deformation region can be done. Perhaps, further analyses of the resistance responses during incremental amplitude cyclic tensile loading may be performed (see e.g. Polymers 2018, 10(7), 777;)
Response: For this initial study, only inelastic deformation was considered however we have addressed the impact of elastic deformation and modified the text accordingly.

Reviewer 2 Report
Reviewer report for carbon-938525
Electrically conductive MXene-coated glass fibers for damage monitoring in fiber-reinforced composites
By Christine B. Hatter, Asia Sarycheva, Ariana Levitt, Babak Anasori, Latha Nataraj, and Yury Gogotsi
This work describes the potential application of electrically conductive MXenes to form composite with glass fibres. A key aspect investigated in this work is electrical resistance for damage monitoring in glass fibre composites or polymer matrix composites (PMC) or fibre reinforced composites (FRP) in general.
Overall, the work shows interesting, new and direct application of MXenes in existing material framework to solve relevant issues in the aerospace and military applications. This paper can be accepted for publication in Journal of carbon research after revision. Some specific comments are included to help authors improve the manuscript:
Specific suggestion:
- This work lacks comparison with existing, carbon based materials that has been widely used in PMC. CNT may be a bit challenging to disperse in polymer systems, but graphene should have less concern. The authors are suggested to include and compare the literature data, both on the aspect of strength enhancement detection (e.g. resistance change). If experimental data is available, it is even better as a fair reference to the MXene. Other factors like higher density/weight for MXene vs. carbon based material should also be included.
- The study seems to be limited to unidirectional composites, and the testing regimen is limited only to (DC) resistance
- Better consideration on real PMC and FRP architecture should also be commented. Apart from unidirectional types, it is known that PMC and FRP structure may also take the form of cross-ply and 2D or 3D woven fibres. The authors are suggested to show, either by measuring or explaining how simple measurements on unidirectional composite can be related to these real FRP architecture. One main reason for this question is due to the known electrical anisptropy (e.g. between the through-plane and in-plane) in FRP. (see e.g. Journal of Composite Materials. 2020;54(7):867-882.)
- While DC measurement is OK for fibre breakage detection (see e.g. Composites: Part B 42 (2011) 77–86), fibre breakage is just one of possible failure modes in FRP. AC is more useful for other types of defects like delamination or debonding (see e.g. Compos Sci Technol 2001;61:855–64.). The authors should discuss what is the potential effect of using MXenes-based composite AC based detection is used, and if this is going to be beneficial or not.
- The measurements shown in Figure 5 focuses only to study inelastic deformation related damage. However, better wear/failure prediction may be done if information of the composites behaviour in the elastic deformation region can be done. Perhaps, further analyses of the resistance responses during incremental amplitude cyclic tensile loading may be performed (see e.g. Polymers 2018, 10(7), 777;)
Author Response

(The authors gave the same response as above.)

Round 2
Reviewer 2 Report
In this revised version, the authors have carefully considered the reviewers' advice and have improved the work, particularly in bringing existing works and alternative detections for damage monitoring in context.
I understand that these are early results, which I believe will lead to many follow up works.